# Hybrid Phylogenetic Model Selection using Deep Learning and High-Performance Computing

Hashara Kumarsinghe[1], Tamara Drucks[2], Thomas Wong[1], Arndt von Haeseler[3,4], and Minh Bui[1]

[1] Australian National University, Australia
[2] Vienna University of Technology, Austria
[3] University of Vienna, Austria
[4] Ludwig Boltzmann Institute for Network Medicine

**Abstract.** Model selection is a fundamental step in phylogenetic inference, aiming to identify the evolutionary model that best fits a given multiple sequence alignment (MSA). Tools such as ModelFinder, implemented in the widely-used IQ-TREE software, rely on maximum-likelihood methods and information criteria (e.g, Akaike or Bayesian) to determine the optimal model. Recently, ModelRevelator introduced a deep neural network that predicts one of six commonly used substitution models and the Gamma-distributed rate heterogeneity model directly from the MSA. While ModelFinder is more accurate, it becomes computationally expensive for large MSAs, whereas ModelRevelator offers faster but less precise.

Here, we present ModelFinder-DL, a hybrid model selection framework that integrates ModelRevelator with ModelFinder. In this framework, neural network predictions guide and constrain likelihood-based evaluation, thereby combining efficiency with robustness. To further enhance computational performance, we integrate OpenMP-based parallelism with ModelFinder-DL, enabling efficient multi-core utilization. In our experiments, ModelFinder-DL achieves up to a $1.4\times$ speed-up over the single-threaded ModelFinder baseline. With four OpenMP threads, ModelFinder-DL attains a $5.3\times$ speedup, compared to $3.8\times$ for four-threaded ModelFinder. This demonstrates the first step towards how deep-learning-guided optimization, combined with OpenMP parallelization, can improve efficiency while maintaining high accuracy.

**Keywords:** Phylogenetic inference · Model selection · Deep learning · Maximum likelihood · OpenMP

## 1 Introduction

Reconstructing evolutionary relationships among organisms, known as phylogenetic inference [6], is central to understanding biodiversity and evolution. Applications range from tracking pathogen evolution (e.g., SARS-CoV-2) to describing species diversification. Among available methods, maximum likelihood [6, 9] offers high accuracy but can be computationally expensive compared to simpler heuristic approaches, for large genomic datasets.

Evolutionary model selection is a critical computational step in all typical phylogenetic analyses. It determines how biological sequences evolve over time and directly

affects both accuracy and computational cost [8]. Choosing an inappropriate model can lead to incorrect tree topologies and misleading biological interpretations [12]. Conversely, overly complex models can overfit the data, increasing runtime without improving accuracy [10]. As genomic datasets continue to grow, the need for fast and accurate model selection becomes increasingly essential.

The widely used IQ-TREE software [9] implements ModelFinder [7] to optimize the parameters of each candidate substitution model, and evaluates model fit using Akaike [2] or Bayesian information criteria [11]. Although IQ-TREE includes efficient likelihood optimization and supports multi-core parallelization via the OpenMP API [5], model selection remains computationally demanding for large datasets.

Recent advances in machine learning have introduced alternatives to determine the best evolutionary model from trained networks [1, 3, 4], offering advantages in computational efficiency and scalability over traditional likelihood-based methods. However, machine learning applicability is limited, as it currently seems impossible to train a network that discovers all available evolutionary models.

To balance speed and accuracy, we propose ModelFinder-DL(MF-DL), a hybrid framework that combines ModelRevelator [4] with IQ-TREE's ModelFinder. MF-DL leverages the computational efficiency of ModelRevelator for DNA substitution and rate-heterogeneity model selection by using the predictions to guide ModelFinder's likelihood evaluation while retaining robustness of the likelihood approach. It supports GPU inference and OpenMP parallelization for scalability across multi-core systems. Our results demonstrate that the MF-DL reduces runtime without compromising phylogenetic accuracy, showcasing the promise of combining deep learning with HPC for large-scale evolutionary studies.

## 2   Methods

Our framework integrates ModelRevelator [4], which includes two neural networks, NNmodelFind and NNalphaFind, into IQ-TREE's ModelFinder [7]. In the standard implementation, ModelFinder evaluates all candidate substitution models to identify the best-fit. In ModelFinder-DL (Fig. 1), NNmodelFind predicts probabilities over six substitution models, and the top-k models with cumulative probability $\geq 95\%$ are evaluated by ModelFinder. This threshold is user-configurable, but we choose $95\%$ as the default because probability-mass filtering in statistical and machine-learning contexts commonly retains 90–97% of the total probability to balance coverage and efficiency. For rate-heterogeneity modeling, NNalphaFind predicts the initial Gamma shape parameter ($\alpha$). Together, these steps accelerate model selection while preserving accuracy. The neural inferences were performed via ONNX Runtime, with the GPU used only for neural network predictions. The implementation remains compatible with IQ-TREE's OpenMP parallelization, enabling efficient multi-core utilization.

Experiments were conducted on the empirical DNA dataset Wu et al. 2018 [13], containing 90 mammal taxa and 15,486 partitions, using the Gadi supercomputer. Since processing the full dataset would require several days, a subset of 500 partitions was randomly selected using a fixed seed to ensure reproducibility and achieve a feasible runtime for benchmarking. Partition merging was performed using IQ-TREE's PartitionFinder module [8], which identifies and merges partitions with similar evolutionary

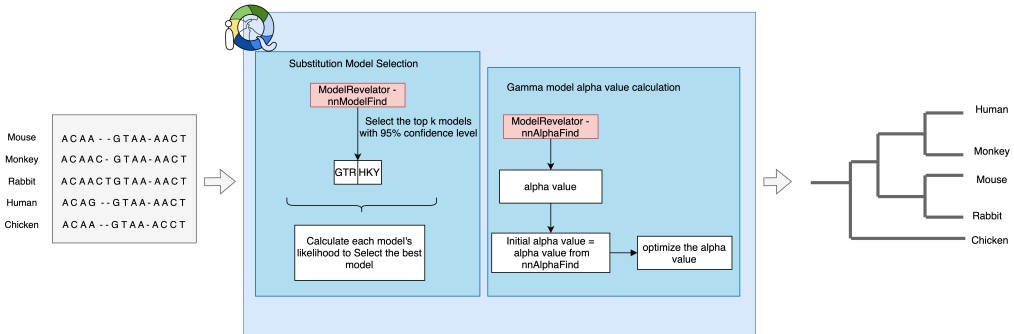

Fig. 1: Hybrid ModelFinder-DL workflow. Neural predictions from ModelRevelator (NN-modelFind, NNalphaFind) guide IQ-TREE's likelihood evaluation, reducing redundant model tests and accelerating selection.

patterns. As NNmodelFind supports six substitution models (GTR, JC, K80, F81, HKY, TN), IQ-TREE was run with the same set for fairness. We benchmarked four configurations: baseline ModelFinder, ModelFinder-DL, and their 4-thread OpenMP variants. All MF-DL experiments were performed on GPU environments.

## 3   Results

For the subsampled Wu et al. 2018 dataset, the baseline ModelFinder took 14.4 h of wall-clock time, whereas ModelFinder-DL took 10.3 h, achieving a 1.4× speed-up. The full dataset (15,486 partitions with 9,150,597 sites, 30× larger) would thus scale to ~432 h and ~309 h, respectively, assuming linear behavior. These estimates indicate a clear advantage of the GPU-accelerated MF-DL for large-scale analyses, although real-world scaling may deviate from linear. With 4 OpenMP threads, the baseline ModelFinder took 3.7 h (3.8x speedup over single-threaded), while the MF-DL took 2.7 h, achieving a 5.3x speedup (Figure 2).

A direct comparison between ModelRevelator and MF-DL on the full dataset is not possible due to IQ-TREE's partition-merging step, which removes correspondence with the original alignment structure required by ModelRevelator. We therefore performed all comparisons on the unmerged subset of 500 partitions. ModelRevelator agreed with ModelFinder on 47.2% of partitions, while MF-DL agreed on 75.6%. This difference reflects their design goals, ModelRevelator provides a single independent prediction and is not intended to match ModelFinder's choices, whereas MF-DL re-evaluates the top-k ModelRevelator candidates using ModelFinder's exact likelihood and AIC/BIC scoring. Consequently, MF-DL attains higher agreement because it preserves ModelFinder's decision rule while operating on a reduced search space.

ModelFinder chieves consistently lower information-criterion scores than MF-DL. Specifically, MF-DL obtains an AIC score of 11,410,020.5450 and a BIC score of 11,420,150.0469, whereas the baseline ModelFinder achieves lower values of 11,406,216.9905 (AIC) and 11,416,961.0436 (BIC). These results indicate that MF attains a better statistical fit, largely because it performs an exhaustive evaluation of all candidate models.

To further assess accuracy, we reconstructed phylogenetic trees using the best-fit models from each configuration. The topological comparison between ModelFinder and

ModelFinder-DL revealed only a minor clade swap, suggesting that the hybrid approach preserves phylogenetic accuracy to a high degree. Future improvements to MF-DL may reduce these discrepancies while maintaining its computational advantages.

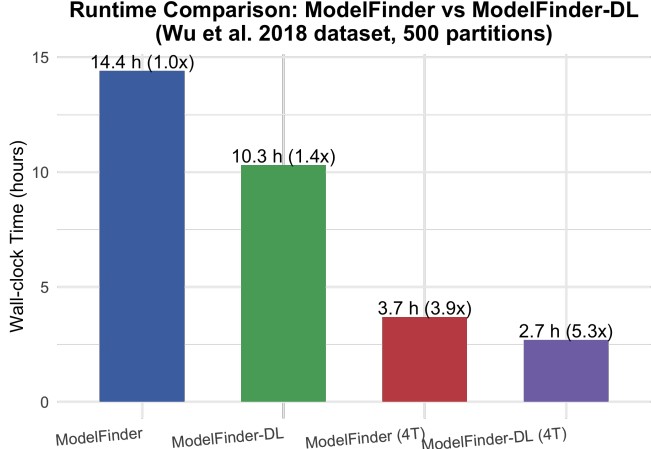

Fig. 2: Wall-clock time comparison on the Wu et al. (2018) dataset. ModelFinder-DL achieves 1.4× (single-thread) and 5.3× (4-thread) speedups over the baseline.

## 4    Conclusion and Future works

This paper presents ModelFinder-DL, a hybrid framework that integrates the deep-learning-based ModelRevelator with the likelihood-based ModelFinder in IQ-TREE. ModelFinder-DL reduces computation time while maintaining comparable phylogenetic accuracy. The OpenMP implementation enables efficient multi-core utilization, achieving speed-up compared to the single-threaded baseline. Our experiments on the Wu et al. (2018) dataset demonstrate up to a 1.4× speed-up in single-threaded runs and a further 5.3× speed-up when executed with OpenMP using four threads. These results are preliminary but highlight the feasibility and promise of combining deep learning with HPC for accelerating large-scale phylogenetic analyses.

In future work, we plan to extend the range of substitution and rate-heterogeneity models supported by the neural networks. This hybrid framework lays the foundation for next-generation phylogenetic tools that effectively leverage both deep learning prediction and likelihood-based methods.

## 5    Acknowledgments

This work was supported by Chan-Zuckerberg Initiative grants for essential open-source software for science (EOSS-0000000132 and EOSS4-0000000312 to B.Q.M.), an Australian Research Council Discovery Project (DP200103151 to B.Q.M.). Computational resources were provided by the Australian Government through the National Computational Infrastructure (NCI) under the ANU Merit Allocation Scheme. H. Kumarasinghe acknowledges support from the Australian Government Research Training Program.

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
