# OpenReview forum: "Hybrid Phylogenetic Model Selection using Deep Learning and High-Performance Computing"
_AJCAI/2025/Workshop/AIML-CEB — AIML-CEB 2025 Poster_

### Official Review · Reviewer_Extk · 2025-11-08
**Hybrid deep learning + likelihood-based evaluation model for evolutionary model prediction from a multiple sequence alignments**

**Rating:** 7
**Confidence:** 2

**Review:**

Various evolutionary models can be predicted from a MSA, IQ-TREE uses ModelFinder and recently ModelRevelator introduced a DL method that predicts 1 of 6 (classifier?) evolutionary models. ModelFinder is more accurate, but more computationally expensive, ModelRevelator is faster but less precise. The authors propose ModelFinder-DL, a hybrid approach: NN predictions guide likelihood-based evaluation. They also employ OpenMP for parallelisation.
Results show some speed improvements while retaining accuracy. They run an experiment on the Wu 2018 dataset, they compare 2 models (ModelFinder, ModelFinder-DL) + multiprocessing variants. Speed: MF 14h, MF-DL 10h and with OpenMP: MF 3.7h, MF-DF 2.7h.

Comments:
* Good introduction to help a reader outside the field
* The focus of the paper seems to be on the speed gain of MF-DL: but it seems like the use of OpenMM/multiprocessing has a much greater influence on speed.
* Performance should also be clearly described: a “minor” difference is mentioned but data not shown: models should be compared on both performance and speed, so performance data should be shown.
* I would have liked to see ModelRevelator being compared in this study, since the MF-DL approach is presented as a hybrid between MF and ModelRevelator.
* I appreciated the inclusion of future work plans.

---

### Official Review · Reviewer_9p84 · 2025-11-10
**Phylogenetic model evaluation using hybrid DL + maximum-likelihood-based approach**

**Rating:** 8
**Confidence:** 3

**Review:**

This paper seeks to address the high computational expense of maximum-likelihood-based methods for phylogenetic inference, by proposing a hybrid deep neural network and maximum-likelihood selection framework. The deep neural network component (ModelRevelator) is used to efficiently reduce the search space of candidate evolutionary models evaluated by the maximum-likelihood-based ModelFinder. The hybrid approach achieves a 1.4-5.3x speedup, with greater efficiency gains achieved through OpenMP parallelisation.

Comments:
- The paper was well-written, with the advances clearly motivated and described.
- The evaluation of phylogenetic accuracy was described very briefly, with details lacking, so it is difficult to assess this component of the study.
- As an additional baseline, the authors should also compare their hybrid framework with the NN-based ModelRevelator (in terms of both speed and accuracy). Currently, they compare only against the maximum-likelihood-based ModelFinder.
- Is there a justification for the filtering threshold of 95% for the ModelRevelator predictions? Could different thresholds be tested for downstream impact on accuracy and efficiency of the hybrid model, to demonstrate the optimality of 95%?
- To strengthen the work for submission to future venues, it could be interesting to assess the relationship between the number of OpenMP threads and efficiency gains across datasets of different sizes, to establish an upper bound on the number of threads to use, to guide the user’s resource allocation.

---

### Official Review · Reviewer_nSTJ · 2025-11-11
**Not sure of the value-add aside from being faster than ModeFinder (but no comparison to ModeRevelator?).**

**Rating:** 6
**Confidence:** 4

**Review:**

It is unclear how much advantage in accuracy or robustness (or speed) ModelFinder-DL provides over ModelRevelator alone, which claims to perform comparably to ModelFinder (but much faster for large alignments). It would also be good to know what the size of the alignment was that the benchmarking was done on to demonstrate the 1.4x and 5.3x speed ups. ModelRevelator reports a 14x speed up over ModelFinder for large MSAs while not suffering any hits to accuracy. Without knowing what else the new model is bringing to the table it is difficult to get excited about parallelising with OpenMP. It would also be good to know whether the speed ups are on the scale of minutes, hours, days etc.

This talk could be interesting to the more generalist (comparatively) audience if there is sufficient background given around what an evolutionary model is and why it is important to get it right...and fast.

---

### Decision · Program_Chairs · 2025-11-12

Accept (Poster)